# Stable Bicyclic Functionalized Nitroxides: The Synthesis of Derivatives of Aza-nortropinone–5-Methyl-3-oxo-6,8-diazabicyclo[3.2.1]-6-octene 8-oxyls

**DOI:** 10.3390/molecules26103050

**Published:** 2021-05-20

**Authors:** Larisa N. Grigor’eva, Alexsei Ya. Tikhonov, Konstantin A. Lomanovich, Dmitrii G. Mazhukin

**Affiliations:** N.N. Vorozhtsov Novosibirsk Institute of Organic Chemistry SB RAS, Academician Lavrentiev Ave. 9, 630090 Novosibirsk, Russia; lng@nioch.nsc.ru (L.N.G.); lomanovich@nioch.nsc.ru (K.A.L.)

**Keywords:** bicyclic nitroxide, condensation, acetylacetone, base-catalyzed recyclization, 3-imidazoline nitroxide, TEMPON, 8-hydroxy-5-methyl-3-oxo-6,8-diazabicyclo[3.2.1]-6-octene

## Abstract

In recent decades, bicyclic nitroxyl radicals have caught chemists’ attention as selective catalysts for the oxidation of alcohols and amines and as additives and mediators in directed C-H oxidative transformations. In this regard, the design and development of synthetic approaches to new functional bicyclic nitroxides is a relevant and important issue. It has been reported that imidazo[1,2-*b*]isoxazoles formed during the condensation of acetylacetone with 2-hydroxyaminooximes having a secondary hydroxyamino group are recyclized under mild basic catalyzed conditions to 8-hydroxy-5-methyl-3-oxo-6,8-diazabicyclo[3.2.1]-6-octenes. The latter, containing a sterically hindered cyclic *N*-hydroxy group, upon oxidation with lead dioxide in acetone, virtually quantitatively form stable nitroxyl bicyclic radicals of a new class, which are derivatives of both 2,2,6,6-tetramethyl-4-oxopiperidine-1-oxyl (TEMPON) and 3-imidazolines.

## 1. Introduction

Among various types of organic radicals, nitroxyl radicals (NRs) are the most stable, most robust, and best-studied class of paramagnetic species. In addition to numerous publications on their synthesis and properties [1,2,3,4,5,6,7,8,9], more and more reports include numerous examples of their use. Thus, NRs are actively employed as oxidants and as trapping reagents in synthetic chemistry [10], as ligands in metal complex catalysis [11], as agents in nitroxide-mediated radical polymerization [12], as molecular spin probes [13], and spin labels [14] and in many other applications [15,16].

In the last decade, new areas of the use of nitroxides have been actively developing. For instance, NRs can participate in UV—or visible-light—excited photosynthesis owing to their ability to participate in light-induced hydrogen atom transfer processes [17,18,19]. A popular NR, 2,2,6,6-tetramethylpiperidine 1-oxyl (TEMPO), has been employed as a mediator for obtaining *N*-centered radicals during electrosynthesis. For example, TEMPO-mediated electrolysis of anilides or ketoximes generates amidyl or iminoxy radicals that trigger subsequent cyclization via an interaction of the forming radical with an unsaturated part of the molecule thereby leading to the closure of a pyrroline-type structure or cyclic nitrone [20]. With a sterically more hindered dispirocyclic NR under the conditions of electrochemical synthesis from sodium azide, it is possible to generate an azidyl radical, which at low temperatures can react twice with alkenes thus implementing alkene diazidation [21]. Taking into account all these “old” and new abilities of stable NRs, it seems worthwhile to direct our efforts toward the design and synthesis of new classes of functional nitroxides that may find applications in various modern fields of materials chemistry and biology.

Bicyclic NRs (BNRs) have been known for a long time, starting from the 1960s (Figure 1, structures **A**–**G**); however, their active use as catalysts for the oxidation of alcohols, in particular, those containing a sterically hindered hydroxyl group, began relatively recently [22,23,24,25,26]. Other functional groups—primary amines [27], diols [28], and cyclic carbamates [29]—as it turned out, are also prone to oxidative transformations in the presence of BNR mediators, forming nitriles and lactone derivatives. Moreover, BNRs were recently shown to be superior additives for enantioselective Ni-catalyzed diarylation of vinylarenes [30] and as mediators for electrochemical α-cyanation of secondary piperidines, morpholine, and pyrrolidine [31].

In the present work, we report the first synthesis of BNRs that are derivatives of 5-methyl-3-oxo-6,8-diazabicyclo[3.2.1]-6-octene 8-oxyl, combining 2,2,6,6-tetramethyl-4-oxopiperidine-1-oxyl (TEMPON) and 3-imidazoline NR moieties (Figure 1, structure **H**).

This paper is a tribute to Professor Leonid B. Volodarsky who was the founder of the school of stable NRs of the imidazoline series and has inspired our research by his brilliant ideas for over 40 years.

## 2. Results

### 2.1. The Synthesis of Bicyclic Nitroxides

Previously, it has been demonstrated that the reaction of acetylacetone with 2-hydroxyamino oximes having a hydroxyamino group at the secondary carbon atom, **1a**–**d**, proceeds through a number of intermediates in tautomeric equilibrium with each other and finally leads to derivatives of tetrahydroimidazo[1,2-*b*]isoxazoles **2a**–**d** [39]. Refluxing of the latter in aqueous alkali is accompanied by the loss of a water molecule followed by recyclization, which entails the opening of the five-membered heterocycle and the formation of a six-membered tetrahydropyridine derivative (cyclic enaminoketone **3a**–**d**) as the main product and 3-acetyl-1-hydroxypyrrole **4a**–**d** as a minor product. We found that this alkali-catalyzed reaction for compounds **2a**,**c**,**d** under much milder conditions (cooling at 0–5 °C, 3–4 h) yields a completely different result: bicycle **5** becomes the main product. Its structure was proved on the basis of spectral and analytical data. For instance, IR spectra of compounds **5a**,**c**,**d** (in KBr) contain intense bands at 1714–1716 and 1629–1639 cm^−1^, corresponding to stretching vibrations of isolated C=O and C=N bonds, as well as a band at 3600 cm^−1^ (in CHCl_3_), corresponding to the presence of an OH group. Compounds **5a**,**c**,**d** do not have absorption maxima in the UV spectrum above 220 nm, indicating the absence of a nitrone group or a conjugated system of unsaturated moieties in the molecule. Analysis of their ^1^H and ^13^C NMR spectra points to the existence of bicycle **5** in the form of a mixture of three isomers, one of which is cyclic hemiacetal **5C** and the other two are the respective hydroxyamino derivatives in which the N-OH bond is oriented either toward the keto group (compound **5O**-***cis***) or away from it (compound **5O**-***trans***) (Scheme 1). Accordingly, in the PMR spectra of compounds **5a**,**c**,**d** in a weak field, three distinct broadened signals are observed (at 7.45–7.54, 7.88–7.92, and 8.29–8.33 ppm), corresponding to protons of the C-OH group of the hemiacetal and N-OH protons for isomers **5O**-***cis*** and **5O**-***trans***. The ratio of these species is shifted toward the prevailing, hypothetically, **5O**-***trans*** conformation (47–65%), while the proportion of **5C** is 22–44%, and the minor component is always **5O**-***cis*** (1–13%). This structural assignment is confirmed by the ^13^C NMR spectra of bicycles **5a**,**c**,**d**, in which carbonyl C atoms for **5O**-***cis*** and **5O**-***trans*** isomers are clearly distinguished at ~206 and 207 ppm, whereas the unique quaternary carbon atom of **5C**, linked to two oxygen atoms, appears at 114 ppm (Appendix A).

We found that bicycle **5**, encumbered with a sterically hindered N-OH group, can be oxidized by lead dioxide to form a stable NR (**6a**, **6c**, or **6d)** as the only reaction product. Acetone is a solvent of choice for the preparation of BNRs; in this case, the yields of compounds **6a**,**c**,**d** could reach 80–87% (Scheme 1). It should be noted that to achieve the highest yield of the radical, pure recrystallized samples of bicycle **5** should be used because the presence of impurities in the substrate affects the stability of **6** during its isolation, namely, during the concentration of the reaction solution. BNRs **6a**,**c**,**d** are solid purple fine-crystalline compounds stable for a long time in a solid state in ambient air at room temperature in the dark. The radicals are readily soluble in polar organic solvents and organochlorides and relatively well soluble in water; for example, the solubility of BNR **6c** at ambient temperature is ~2.5 × 10^−2^ M. Their IR spectra resemble those of their diamagnetic precursors, except that the band at >3100 cm^−1^ is absent due to vibrations of the N-OH group.

In the mass spectra of BNRs **6**, along with the molecular ion peak at 181 (**6a**), 207 (**6c**), or 221 (**6d**), M + 1 ions are present, and the most intense peak belongs to the M − 30 ion corresponding to a loss of the N-O group (Appendix A).

To confirm the structure of the obtained BNRs **6**, they were reduced to their diamagnetic precursors **5** via a reaction with zinc powder in the presence of NH_4_Cl in aqueous acetone.

### 2.2. EPR Study of New Bicyclic Nitroxides ***6***

The EPR spectra of BNRs **6** recorded for degassed solutions in toluene represent triplets with hyperfine coupling (*hfc)* constant a_N_ of ~1.855 mT, for one of which (radical **6c**), a fine structure is observed due to the presence of a hyperfine interaction with hydrogen atoms in the CH_2_^−^ and CH_3_^−^ groups (Figure 2).

A similar splitting of the triplet has been documented earlier by Rassat et al. for the NR series of nortropinone (Figure 1, structure **D**), in which the hydrogen atoms located on the α- and α′-carbon atoms of the carbonyl group (C-2,4) interact with an unpaired electron, with the a_H_ constant of 0.25 mT (2H) or 0.12 mT (2H) depending on whether they are in the axial or equatorial position of the six-membered ring [35]. In our case, due to the asymmetry of all carbon atoms in **6c**, the *hfc* constants for all the different types of hydrogen atoms differ from each other and are in the range from 0.095 to 0.217 mT for methylene protons in the carbonyl and decrease from 0.054 mT for protons of the CH_2_ group on the six-membered ring closest to the radical center down to ~0.001 mT for the most distant protons of the aliphatic ring (Appendix A).

To the best of our knowledge, such BNRs containing both structural fragments of 2,5-dihydroimidazole and 4-piperidone have not been described previously in the literature. There is only one known example of the formation of bicyclic 1-hydroxy-3-imidazoline 3-oxide **7**, which is carried out via the cyclodimerization of 3-methylbut-3-en-2-one oxime **8** with a low yield upon extra-long heating in a sealed ampoule, presumably through tetrahydropyridine intermediate **9** (Scheme 2) [40]. Although the putative structure of compound **7** has been confirmed by X-ray diffraction data [41], later, German researchers found that most of the reactions of bicyclic **7** proceed through the monocyclic tautomeric form of methylnitrone **10** [42]. They also showed that during the oxidation of solutions of *N*-hydroxy derivative **7** in various polar and nonpolar solvents by Ag_2_O, PbO_2_, hydrogen peroxide, or *tert*-butoxy or benzoyloxy radicals, the corresponding BNR **11** is generated with a typical a_N_ of 1.9 mT and an isotropic g-factor (g_iso_) of 2.0058–2.0066 [43]. Nevertheless, the publications just cited do not say how stable the nitroxide **11** prepared in an EPR spectrometer ampoule is.

According to our observations and similar data in the literature, now we can propose a mechanism of base-catalyzed recyclization of imidazoisoxazoles **2**. The mechanism consists of the transformation of **2** into an intermediate (2*H*-imidazole *N*-oxide **2*H***-**Im**; Scheme 1), whose subsequent intramolecular interaction with the enolate anion generated in the reaction can occur both at the carbon atom of the nitrone group and at the carbon atom of the imino group, thus leading either to the product of kinetic control (bicyclic hydroxylamine **5** in the former case) or to the product of thermodynamic control (4-pyridone-2-ketoxime **3** in the latter case). Indeed, in the control experiment, when we introduced a crystallized sample of bicycle **5d** into the reaction, with short heating (60–65 °C) of this aqueous solution in 0.25N NaOH, **5d** virtually quantitatively turned into a single product, pyridone **3d**, which was isolated and characterized by NMR spectroscopy.

## 3. Materials and Methods

### 3.1. General Information

Fourier transform infrared spectra (FT-IR) were recorded in KBr pellets on a Bruker Vector-22. UV-Vis spectra were obtained for EtOH solutions of bicyclic nitroxides **6a**,**c**,**d** using a Hewlett-Packard HP 8453 spectrophotometer. ^1^H NMR and ^13^C NMR spectra were recorded on Bruker AV-300, AV-400, DRX-500, and AV-600 spectrometers at 300/400/500/600 and 75/100/125/150 MHz, respectively, for 3–10% solutions of compounds **2**, **3**, and **5** in DMSO-*d*_6_; the positions of signals were determined relative to a residual proton signal [DMSO-*d*_6_ (2.50 ppm) for ^1^H spectra] or a carbon signal [DMSO-*d*_6_ (39.4 ppm) for ^13^C spectra] of the deuterated solvent. The analyses were performed on a Thermo Scientific DFS (Double Focusing System) high-resolution mass spectrometer (Thermo Electron Corp., Waltham, MA, USA); the recording mode was electron ionization with an ionizing electron energy of 70 eV, and exact mass was measured relative to the lines of the standard perfluorokerosene.

Elemental analyses were carried out on a Euro EA 3000 automatic CHNS analyzer. Melting points were determined by means of an FP 81 HT instrument (Mettler Toledo, Columbus, OH, USA). Column chromatography and thin-layer chromatography (TLC) were conducted using Acros silica gel 60A (0.035–0.070 mm) and Sorbfil PTLC-AF-UV 254 (Imid, Krasnodar, Russia), respectively, with eluents EtOAc and CHCl_3_-MeOH.

EPR spectra of nitroxides **6a**,**c**,**d** were acquired by means of a Bruker Elexsys E540 X-band continuous-wave EPR spectrometer at 295 K for diluted (10^−4^ M) and oxygen-free toluene solutions. Experimental settings for **6c** were as follows: microwave power, 0.63 mW; modulation frequency, 100 kHz; modulation amplitude, 0.002 mT; the number of points, 2048; sweep magnetic field, 6 mT; the number of scans, 160; and the time constant, 20 ms. Experimental settings for **6a**,**d** were as follows: microwave power, 2.0 mW; modulation frequency, 100 kHz; modulation amplitude, 0.05 mT; the number of points, 1024; sweep magnetic field, 10 mT; the number of scans, 8; and the time constant, 20 ms. The following experimental settings for the center line of **6c** were applied: microwave power, 0.63 mW; modulation frequency, 100 kHz; modulation amplitude, 0.002 mT; the number of points, 1048; sweep magnetic field, 1.5 mT; the number of scans, 196; and the time constant, 20 ms. To determine g_iso_ values, X-band continuous-wave EPR spectra of a mixture of the analyzed radical (**6a**, **6c**, or **6d**) with TEMPO were recorded. Then, the known g_iso_ of TEMPO was used for a spectrum simulation, and the target g_iso_ value was excluded. Simulations of solution EPR lines were carried out in the Easy Spin software, which is available at www.easyspin.org (accessed on 18 May 2021).

### 3.2. Synthesis

#### 3.2.1. Imidazo[1,2-*b*]isoxazoles **2a**,**c**,**d**

Compounds **2a**,**c**,**d** were synthesized via a reaction of 2-hydroxyamino oximes **1a**,**c**,**d** [44] with acetylacetone [39]. Their ^1^H NMR spectra and other data are consistent with those described in the literature.

*6-Hydroxy-2,3,6,7a-tetramethyl-3,6,7,7a-tetrahydroimidazo[1,2-b]isoxazole 1-oxide* (**2a**) (a mixture of three diastereomers). ^13^C NMR (150 MHz, DMSO-*d*_6_), δ, ppm: 9.8, 10.17*, 10.2 (C(2)-*C*H_3_); 12.2, 16.8*, 18.5 (C(3)-*C*H_3_); 19.2, 23.7*, 24.7, 24.8, 25.3*, 25.9 (C(7a)-*C*H_3_ and C(6)-*C*H_3_); 48.9, 49.6, 50.6* (C-7); 62.2, 66.3*, 70.5 (C-3); 98.3, 98.9*, 99.2 (C-7a); 100.4, 101.3*, 104.9 (C-6); 137.1, 138.3*, 139.3 (C-2). (The asterisk denotes a minor isomer).

*2-Hydroxy-2,3a-dimethyl-2,3,3a,5,6,7,8,8a-octahydrobenzo[d]isoxazolo[2,3-a]imidazole 4-oxide* (**2c**) (a mixture of two diastereomers). ^13^C NMR (75 MHz, DMSO-*d*_6_), δ, ppm: 23.6, 23.7*, 23.8*, 24.7, 25.2* (C-6, C-7, C-8); 24.7*, 24.8, 25.1*, 25.6 (C(3a)-*C*H_3_ and C(2)-*C*H_3_); 32.5*, 34.6 (C-5); 48.8, 50.4* (C-3); 69.0*, 72.7 (C-8a); 99.6, 100.0* (C-3a); 102.2*, 106.0 (C-2); 140.9*, 141.7 (C-4a); (The asterisk denotes a minor isomer).

*2-Hydroxy-2,3a-dimethyl-3,3a,5,6,7,8,9,9a-octahydro-2H-cyclohepta[d]isoxazolo[2,3-a]imidazole 4-oxide* (**2d**) (a mixture of six diastereomers). ^13^C NMR (150 MHz, DMSO-*d*_6_), δ, ppm: 12.8, 13.5, 18.6*, 18.7, 19.9, 23.6, 24.7, 24.9, 25.3, 25.4, 25.96, 26.03* (C(3a)-*C*H_3_ and C(2)-*C*H_3_); 24.0, 24.2, 24.3, 24.5*, 24.7, 25.5, 26.6*, 26.8, 26.9, 27.0, 27.7, 27.8, 27.9, 27.94, 28.7, 28.8, 28.9*, 29.2, 30.5*, 30.53, 30.58, 33.9, 34.1, 34.8, 36.4* (C-5, C-6, C-7, C-8, C-9); 48.7, 49.1, 49.8*, 50.6, 56.3 (C-3); 66.7, 68.1*, 69.6, 71.2, 75.4, 77.8 (C-9a); 98.3, 98.7, 99.0, 99.6*, 100.7*, 101.7, 102.8, 105.4, 105.9, 106.8 (C-3a and C-2); 141.2, 141.6, 142.8*, 143.7, 144.2, 145.3 (C-4a); (The asterisk denotes the main isomer).

#### 3.2.2. The General Synthetic Procedure for the Recyclization of Imidazo[1,2-*b*]isoxazoles **2a**,**c**,**d** to 8-Hydroxy-5-methyl-3-oxo-6,8-diazabicyclo[3,2,1]-6-octenes **5a**,**c**,**d**

To a cold solution (0–5 °C) of imidazoisoxazole **2** (10 mmol) in 10–15 mL of water, a solution of 10 mmol of NaOH in 14 mL of water was added with stirring for 20 min and was kept on ice for 3 h. The reaction mixture was neutralized with 10% hydrochloric acid in an ice-water bath, and NaCl was added until saturation at room temperature. The precipitate of bicycle **5** was filtered off, washed with a small amount of ice-cold water, and dried until constant weight.

*8-Hydroxy-1,5,7-trimethyl-6,8-diazabicyclo[3.2.1]oct-6-en-3-one* (**5a**), colorless crystals, isolated yield 0.72 g (40%), m.p. 175–177 °C (*i*-PrOH). Elemental analysis: found: C, 59.42; H, 7.90; N, 15.45; calcd. for C_9_H_14_N_2_O_2_: C, 59.32; H, 7.74; N, 15.37%. IR spectrum, ν, cm^−1^, (KBr): 3124, 2978, 2847, 1714 (C=O), 1637 (C=N), 1456, 1375, 1315, 1242. ^1^H NMR (400 MHz, DMSO-d_6_), δ, ppm (*J*, Hz): A: Main isomer (keto form): 1.16 (s, 3H, CH_3_); 1.29 (s, 3H, CH_3_); 1.87 (s, 3H, CH_3_); 1.95 (d, *J* = 4.4, 1H, CH); 1.99 (d, *J* = 4.4, 1H, CH); 2.47 (d, *J* = 17.0, 1H, CH); 2.57 (d, *J* = 17.0, 1H, CH); 8.33 (brs, 1H, N-OH). B: Cyclic hemiacetal: 1.21 (s, 3H, CH_3_); 1.34 (s, 3H, CH_3_); 1.90 (s, 3H, CH_3_); 1.49 (dd, *J* = 2.8, 11.8, 1H, CH); 1.57 (d, *J* = 11.8, 1H, CH); 1.67 (dd, *J* = 2.8, 11.8, 1H, CH); 1.80 (d, *J* = 11.8, 1H, CH); 7.45 (brs, 1H, C-OH). C: Minor isomer (keto form): 1.29 (s, 3H, CH_3_); 1.41 (s, 3H, CH_3_); 1.91 (s, 3H, CH_3_); 2.29 (dd, *J* = 2.1, 17.3, 1H, CH); 2.43 (dd, *J* = 2.1, 17.3, 1H, CH); 7.88 (brs, 1H, N-OH). ^13^C NMR (100 MHz, DMSO-d_6_), δ, ppm: A: Main isomer (keto form): 16.4 (CH_3_); 18.8 (CH_3_); 23.1 (5-CH_3_); 40.7 (CH_2_); 42.0 (CH_2_); 69.8 (C-1); 85.8 (C-5); 177.8 (C=N); 206.9 (C=O). B: Cyclic hemiacetal: 16.7 (CH_3_); 21.5 (CH_3_); 25.8 (5-CH_3_); 44.5 (CH_2_); 45.9 (CH_2_); 76.1 (C-1); 93.6 (C-5); 114.0 (NOCO); 177.3 (C=N). C: Minor isomer (keto form): 16.4 (CH_3_); 17.1 (CH_3_); 21.3 (CH_3_); 49.0 (CH_2_); 50.3 (CH_2_); 75.1 (C-1); 90.6 (C-5); 175.1 (C=N); 205.8 (C=O).

*8-Hydroxy-5-methyl-1,7-tetramethylene-6,8-diazabicyclo[3.2.1]oct-6-en-3-one* (**5c**), colorless rhombic crystals, isolated yield 1.04 g (50%), m.p. 188–190 °C (abs EtOH). Elemental analysis: found: C, 63.50; H, 7.90; N, 13.33; calcd. for C_11_H_16_N_2_O_2_: C, 63.44; H, 7.74; N, 13.45%. IR spectrum, ν, cm^−1^, (KBr): 3402, 3099, 2941, 2862, 1715 (C=O), 1639 (C=N), 1452, 1313, 1238, 1009, 955, 812. ^1^H NMR (600 MHz, DMSO-d_6_), δ, ppm (*J*, Hz): A: Main isomer (keto form): 1.25–1.87 (m, 6H, (CH_2_)_3_); 1.30 (s, 3H, Me); 1.91 (d, *J* = 12.0, 1H, CH); 1.97 (d, *J* = 12.0, 1H, CH); 2.14–2.70 (m, 2H, 2CH); 2.49 (d, *J* = 12.0, 1H, CH); 2.56 (d, *J* = 12.0, 1H, CH); 8.29 (brs, 1H, N-OH). B: Cyclic hemiacetal: 1.25–1.87 (m, 10H, (CH_2_)_3_ and CH_2_COCH_2_); 1.37 (s, 3H, Me); 2.12–2.70 (m, 2H, 2CH); 7.54 (s, 1H, C-OH). C: Minor isomer (keto form): 1.25–1.87 (m, 6H, (CH_2_)_3_); 1.45 (s, 3H, CH_3_); 2.14–2.32 (m, 3H, 3CH); 2.43 (d, *J* = 18.0, 1H, CH); 2.57 (d, *J* = 18.0, 1H, CH); 2.60–2.70 (m, 1H, CH); 7.89 (brs, 1H, N-OH). D: tetrahydropyridone **3c** (as an impurity): 1.25–2.25 (m, 8H, (CH_2_)_4_); 1.88 (s, 3H, CH_3_); 2.16 (d, *J_AB_* = 18.0, 1H, CH); 2.49 (d, *J_AB_* = 18.0, 1H, CH); 4.57 (s, 1H, CH=); 7.27 (s, 1H, NH); 10.75 (s, 1H, =NOH). ^13^C NMR (150 MHz, DMSO-*d*_6_), δ, ppm: A: Main isomer (keto form): 23.0 (Me); 21.0, 24.3; 29.2, 34.5 ((CH_2_)_4_); 42.5, 46.2 (*C*H_2_-C=O); 68.9 (N-C-C=N); 85.9 (N-C-N); 178.7 (C=N); 207.0 (C=O). B: Cyclic hemiacetal: 25.7 (CH_3_); 21.6, 24.2, 29.7, 36.9 ((CH_2_)_4_); 44.0, 44.8 (*C*H_2_-C=O); 76.2 (N-C-C=N); 94.2 (N-C-N); 114.1 (NOCO); 178.2 (C=N). C: Minor isomer (keto form): 21.5 (Me); 21.2, 24.9, 28.2, 29.0 ((CH_2_)_4_); 48.2, 50.3 (*C*H_2_-C=O); 74.9 (N-C-C=N); 91.0 (N-C-N); 175.7 (C=N); 205.8 (C=O). D: tetrahydropyridone **3c** (as an impurity): 20.6 (CH_3_); 20.1, 20.4, 25.4, 38.2 ((CH_2_)_4_); 44.8 (C-5); 58.6 (C-6); 96.9 (C-3); 156.9, 160.4 (C-2 and C-7); 190.0 (C-4).

*8-Hydroxy-5-methyl-1,7-pentamethylene-6,8-diazabicyclo[3.2.1]oct-6-en-3-one* (**5d**), colorless rhombic crystals, isolated yield 1.11 g (50%), m.p. 203–204 °C (abs EtOH). Elemental analysis: found: C, 65.10; H, 8.31; N, 12.62; calcd. for C_12_H_18_N_2_O_2_: C, 64.84; H, 8.16; N, 12.60%. IR spectrum, ν, cm^−1^, (KBr): 3400, 3093, 2926, 2852, 1716 (C=O), 1630 (C=N), 1448, 1313, 1182, 991, 825. ^1^H NMR (400 MHz, DMSO-d_6_), δ, ppm (*J*, Hz): A: Main isomer (keto form): 1.06–1.90 (m, 8H, (CH_2_)_4_); 1.30 (s, 3H, Me); 1.94 (d, *J* = 16.0, 1H, CH); 2.00 (d, *J* = 16.0, 1H, CH); 2.17–2.42 (m, 2H, 2CH); 2.46 (d, *J* = 16.0, 1H, CH); 2.61 (d, *J* = 16.0, 1H, CH); 8.28 (brs, 1H, N-OH). B: Cyclic hemiacetal: 1.06–1.90 (m, 12H, (CH_2_)_4_ and CH_2_COCH_2_); 1.34 (s, 3H, Me); 2.17–2.42 (m, 2H, 2CH); 7.51 (brs, 1H, C-OH). ^13^C NMR (100 MHz, DMSO-d_6_), δ, ppm: A: Main isomer (keto form): 23.4 (CH_3_); 23.4, 28.4, 30.7, 32.0, 32.1 ((CH_2_)_5_); 41.2, 42.8 (*C*H_2_-C=O); 73.5 (N-*C*-C=N); 85.9 (N-C-N); 182.2 (C=N); 207.3 (C=O). B: Cyclic hemiacetal: 26.0 (CH_3_); 24.3, 27.8, 32.1, 32.5, 35.4 ((CH_2_)_5_); 44.5, 46.2 (*C*H_2_-C=O); 80.1 (N-*C*-C=N); 93.8 (N-C-N); 113.9 (NOCO); 181.9 (C=N).

#### 3.2.3. The General Procedure for the Oxidation of 8-Hydroxy-6,8-diazabicyclo[3.2.1]oct-6-en-3-ones **5a**,**c**,**d**

An amount of 8-hydroxydiazabicyclooctene **5** (0.55 mmol) was stirred in 10 mL of acetone for 5 min; after that, 2.0 mmol of PbO_2_ was added, and the suspension was stirred for 3 h at room temperature and passed through a fine pore glass filter, and the precipitate was washed thoroughly with Me_2_CO (3 × 5 mL). Combined filtrates were evaporated, and the solid residue was diluted with 2 mL of EtOAc and flash chromatographed with an ethyl acetate on a column with silica to obtain a colored fraction containing radical **6**. The solvent was evaporated, the residue was treated with cold hexane, and the precipitate of NR **6** was filtered off.

*1,5,7-Trimethyl-6,8-diazabicyclo[3.2.1]oct-6-en-3-one 8-oxyl* (**6a**). Light purple fine crystals, isolated yield 81 mg (81%), m.p. 66–68 °C (hexane), *R_f_* 0.45 (EtOAc). Elemental analysis: found: C, 59.42; H, 7.60; N, 15.41; calcd. for C_9_H_13_N_2_O_2_: C, 59.65; H, 7.23; N, 15.46%. IR spectrum, ν, cm^−1^, (KBr): 2991, 2937, 2893, 1718 (C=O), 1626 (C=N), 1450, 1371, 1313, 1236. UV (EtOH), λ_max_ nm, (lg ε): 298 (2.65), 312 (2.65). High-resolution mass spectrometry (HRMS), electrospray ionization (EI): Observed 181.0970; C_9_H_13_N_2_O_2_ ([M]^+^); calculated 181.0972. ESR (PhMe): br. triplet; a_N1_ = 1.852 mT; g_iso_ = 2.0054.

*5-Methyl-1,7-tetramethylene-6,8-diazabicyclo[3.2.1]oct-6-en-3-one 8-oxyl* (**6c**). Purple needle crystals, isolated yield 92 mg (81%), m.p. 102–104 °C (hexane), *R_f_* 0.45 (EtOAc). Elemental analysis: found: C, 63.55; H, 7.31; N, 13.30; calcd. for C_11_H_15_N_2_O_2_: C, 63.75; H, 7.30; N, 13.52%. IR spectrum, ν, cm^−1^, (KBr): 2951, 2862, 1716 (C=O), 1628 (C=N), 1448, 1392, 1373, 1317, 1238, 1167. UV (EtOH), λ_max_ nm, (lg ε): 235 (3.30). HRMS (EI): Observed 207.1126; C_11_H_15_N_2_O_2_ ([M]^+^); calculated 207.1126. ESR (PhMe): a_N1_ = 1.855 mT; a_H1_ = 0.217 mT; a_H2_ = 0.1798 mT; a_H3_ = 0.119 mT; a_H4_ = 0.095 mT; a_H5,H6,H7_ = 0.041 mT; a_H8_ = 0.054 mT; a_H9_ = 0.025 mT; a_H10_ = 0.022 mT; a_H11_ = 0.019 mT; a_H12_ = 0.015 mT; a_N2_ = 0.009 mT; a_H13_ = 0.004 mT; a_H14_ = 0.002 mT; a_H15_ = 0.001 mT; g_iso_ = 2.00664.

*5-Methyl-1,7-pentamethylene-6,8-diazabicyclo[3.2.1]oct-6-en-3-one-8-oxyl (***6d**). Dark purple needles, isolated yield 106 mg (87%), m.p. 98–100 °C (hexane), *R_f_* 0.45 (EtOAc). Elemental analysis: found: C, 65.22; H, 8.01; N, 12.57; calcd. for C_12_H_17_N_2_O_2_: C, 65.14; H, 7.74; N, 12.66%. IR spectrum, ν, cm^−1^, (KBr): 2941, 2929, 2856, 1722 and 1711 (C=O), 1616 (C=N), 1446, 1383, 1313, 1317, 1227, 1182. UV (EtOH), λ_max_ nm, (lg ε): 238 (3.30). HRMS (EI): Observed 221.1286; C_12_H_17_N_2_O_2_ ([M]^+^); calculated 221.1285. ESR (PhMe): br. triplet; a_N1_ = 1.841 mT; g_iso_ = 2.00667.

#### 3.2.4. The General Procedure for the Reduction of NRs **6a**,**c**,**d**

To a stirred solution of nitroxide **6** (1.89 mmol) in 15 mL of acetone, zinc powder (123 mg, 1.89 mmol) was added, followed by dropwise addition of a solution of NH_4_Cl (158 mg, 2.27 mmol) in 1 mL of water during 15 min. The mixture was incubated for 25 min, the precipitate was filtered off, and the filtrate was evaporated. The residue was treated with cold water, and the precipitate was filtered off, washed with a minimal amount of water and diethyl ether, and dried until constant weight. The yield of bicyclic hydroxylamine **5** depends on its solubility in water and varies from 48% (for compound **5a**) up to 79% (for **5d**). IR spectra of the obtained compounds are consistent with the IR spectra of those synthesized through the recyclization of **2**.

#### 3.2.5. Base-Promoted Recyclization of 8-Hydroxy-6,8-diazabicyclo[3.2.1]oct-6-en-3-one **5d** into Tetrahydropyridine-4-one **3d**

A solution of 21 mg (0.525 mmol) of NaOH in 1 mL of water was added to a suspension of 111 mg (0.5 mmol) of bicycle **5d** in 1 mL of water, and the resulting solution was stirred at 65 °C for 150 min. The reaction mixture was cooled to ambient temperature, neutralized with 10% aq HCl and kept at 3 °C for 15 h. The precipitate of **3d** was filtered off, washed with 1 mL of ice water, and dried to constant weight. The aqueous filtrate was saturated with NaCl and kept at room temperature for 2 d; the precipitate was filtered off and washed with a minimal volume of ice water to isolate an additional amount of **3d**.

*7-(Hydroxyimino)-2-methyl-1-azaspiro[5.6]dodec-2-en-4-one* (**3d**). Pale fine crystal powder, total isolated yield 84 mg (76%). Its IR and ^1^H NMR spectra are consistent with those described in the literature [39]. ^13^C NMR (75 MHz, DMSO-*d*_6_), δ, ppm: 20.5 (CH_3_); 22.9 (C-11); 23.3 (C-8); 25.7 (C-9); 30.3 (C-10); 39.3 (C-12); 44.9 (C-5); 61.5 (C-6); 96.8 (C-3); 159.2, 159.8 (C-2 and C-7); 190.2 (C-4).

## 4. Conclusions

It was shown in this work that imidazo[1,2-*b*]isoxazoles formed by the reaction of acetylacetone with 2-hydroxyaminooximes having a secondary hydroxyamino group are key compounds for the synthesis of sterically hindered hydroxylamines, precursors of the corresponding BNRs containing two functional groups in a paramagnetic molecule. It should be pointed out that these nitroxides simultaneously contain TEMPON and 3-imidazoline frames in the molecular core; this dualism may open up new possibilities for the synthesis of functional DNP agents, radical polymerization mediators, catalysts of one-electron oxidation, spin-labeled chelating ligands, and other constructs. Being aza-analogs of NRs of the nortropinone series, radicals **6** can become convenient starting compounds for the synthesis of polyfunctional hydrophilic nitroxides of the 3-imidazoline series (cf. [45]).

Considering the increased interest of researchers in BNRs as promising agents for the oxidation of various alcohols, amines, and related compounds (including sterically hindered ones), further work is being planned regarding the synthesis, reactivity evaluation, transformation of the keto function, and determination of electrochemical characteristics of BNRs of the 6,8-diazabicyclo [3.2.1]-6-octene 8-oxyls series.

## Data Availability

Not applicable.

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
