# Peer review of "Stable Bicyclic Functionalized Nitroxides: The Synthesis of Derivatives of Aza-nortropinone–5-Methyl-3-oxo-6,8-diazabicyclo[3.2.1]-6-octene 8-oxyls"

_molecules, 2021, doi:10.3390/molecules26103050_

Round 1
Reviewer 1 Report
This manuscript contributed by Tikhonov, Mazhukin, and co-workers describes the synthesis of a new class of nitroxyl radicals, namely, 5-methyl-3-oxo-6,8-diazabicyclo[3.2.1]-6-octene 8-oxyls. The structure of the nitroxyl radicals and their hydroxylamine precursors have been carefully elucidated based on sufficient spectral data. The nitroxyl radicals containing unique 3-imidazoline structure are expected to have unique properties, encouraging many readers to employ the new radicals. Basically, this referee supports its publication in Molecules provided that the following issues are suitably addressed.
(1) The main topic of this manuscript is the synthesis of new nitroxyl radicals. To explain the novelty of this work, the authors describe nitroxyl radicals previously reported in Figure 1. However, they do not cite any references for the synthesis of each nitroxyl radical. The author should add appropriate references.
(2) The authors emphasize the application of nitroxyl radicals to agents for alcohol oxidation. Thus, it is preferable to describe the preliminary information of the redox properties (e.g. cyclic voltammetry and examination of alcohol oxidation) of the new nitroxyl radicals is added.
Minor points:
(3) Abstract and p2: The full name of TEMPON should be not “4-oxopiperidine-1-oxyl” but “2,2,6,6-tetramethyl-4-oxopiperidine-1-oxyl”
(4) Scheme 1: The green curly arrows indicating the movement of electron pairs should be not half-arrows, but full-arrows.
Reviewer 2 Report
The method of the synthesis and confirmation the structures of new bicyclic nitroxyl radicals is described in the presented manuscript. In my opinion, the paper requires two comments.
1. The three step synthesis of the final nitroxyl radicals 6 is complex and not obvious. The structures of the described compounds 2-> -> 6 are strongly dependent on the complex spectral analysis of a large number of tautomeric and diastereoisomeric isomers. The obtained nitroxyl radicals 6 are crystalline substances. In order to prove finally the structures 6, it would be advisable in future to confirm their structures by X-ray analysis.
2. There is no clear description of the differences in experimental steps for:
2 --> 3,4
2---> 5
a) As shown In Scheme 1, the difference in experimental details between the 2 -> 3.4 and 2 ---> 5 is caused by using of KOH and NaOH, respectively.
b) In the text the difference in experimental details between the 2 -> 3.4 and 2 ---> 5 is presented in a slightly different way
2 -> 3,4: aqueous alkali,
2 ---> 5: using much milder conditions.
c) In the experimental part, however, I cannot find a protocol for the step 2 -> 3,4.
The differences between steps 2 -> 3,4 and 2 ---> 5 should be clearly and uniformly explained both in the text and in Scheme 1. A separate procedure for 2 -> 3,4 should be placed In the experimental part.
